# Gestalt Vision: A Dataset for Evaluating Gestalt Principles in Visual Perception

Jingyuan Sha                                      JINGYUAN.SHA@TU-DARMSTADT.DE
Hikaru Shindo                                     HIKARU.SHINDO@TU-DARMSTADT.DE
Kristian Kersting                                      KERSTING@TU-DARMSTADT.DE
Devendra Singh Dhami                                            D.S.DHAMI@TUE.NL

**Editors:** Leilani H. Gilpin, Eleonora Giunchiglia, Pascal Hitzler, and Emile van Krieken

## Abstract

Gestalt principles, established in the 1920s, describe how humans perceive individual elements as cohesive wholes. These principles, including proximity, similarity, closure, continuity, and symmetry, play a fundamental role in human perception, enabling structured visual interpretation. Despite their significance, existing AI benchmarks fail to assess models' ability to infer patterns at the group level, where multiple objects following the same Gestalt principle are considered as a group using these principles. To address this gap, we introduce Gestalt Vision, a diagnostic framework designed to evaluate AI models' ability to not only identify groups within patterns but also reason about the underlying logical rules governing these patterns. Gestalt Vision provides structured visual tasks and baseline evaluations spanning neural, symbolic, and neural-symbolic approaches, uncovering key limitations in current models' ability to perform human-like visual cognition. Our findings emphasize the necessity of incorporating richer perceptual mechanisms into AI reasoning frameworks. By bridging the gap between human perception and computational models, Gestalt Vision offers a crucial step toward developing AI systems with improved perceptual organization and visual reasoning capabilities.

## 1. Introduction

Gestalt principles—such as proximity, similarity, closure, and continuity—describe the innate ways in which human perception organizes visual information into coherent wholes (Wertheimer, 1938; Koffka, 1935; Ellis, 1999; Palmer, 1999). These principles allow humans to instinctively identify salient features and abstract high-level concepts from complex scenes. For example, we instinctively perceive symmetrical arrangements as unified structures and tend to complete incomplete shapes through closure, enabling rapid recognition of objects and their interrelationships (see Fig. 1). This perceptual strategy is particularly relevant in complex visual reasoning tasks, where it is important to move beyond the focus on individual pixels or discrete objects to discern overarching patterns and structures. Incorporating Gestalt principles enables neuro-symbolic models to better emulate human perception, improving object relationships and high-level reasoning.

Neuro-symbolic systems often combine deep learning models like Mask R-CNN (He et al., 2017) or Slot Attention (Locatello et al., 2020) to detect objects and assign symbolic labels and bounding boxes. These symbolic abstractions form the input to reasoning modules that operate over object-level representations. However, this pipeline tends to overlook

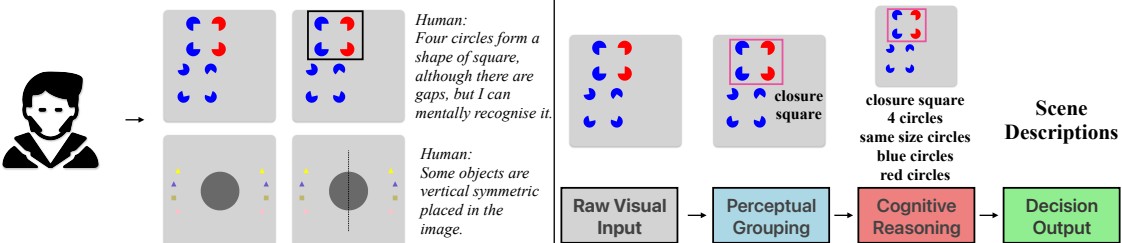

Figure 1: **Gestalt Cognitive Reasoning. Left**: Humans perceive structured patterns through Gestalt principles like closure and symmetry (Koffka, 1935). **Right**: The stages of human perceptual reasoning following Gestalt principles. The process begins with Raw Visual Input.Through Perceptual Grouping, Gestalt principles organize the perception into meaningful structures. This leads to Cognitive Reasoning, where humans interpret these organized patterns. Finally, the Decision Output represents the inferred conclusions drawn from the visual data.

crucial low-level attributes—such as contours, size, color, and shading—that are essential for context-sensitive inference. As a result, reasoning models may miss the nuanced information needed for complex relational or group-level understanding. To overcome this, neuro-symbolic systems must adopt more robust perception mechanisms that preserve both local and global visual features.

To bridge this gap, we introduce the Gesalt Vision (ELVIS), a synthetic benchmark designed to evaluate models' ability to perceive and reason over Gestalt-based groupings and visual rules. Each task in ELVIS is constructed that emphasizes a specific Gestalt principle, with structured visual scenes and rule-based labels. Unlike conventional visual benchmarks, ELVIS focuses on both group-level regularities and isolated object features.

Overall, we make the following contributions:

1. We introduce Gestalt Vision (ELVIS)[1], a new benchmark dataset, covering a broad range of gestalt principles in synthetic scenes.

2. We formalize tasks explicitly constructed to test neuro-symbolic models' reasoning abilities across shape, size, color, and spatial configurations.

3. We evaluate and analyze multiple baseline models, highlighting both the strengths and challenges of existing neuro-symbolic approaches in capturing nuanced perceptual grouping phenomena.

To this end, we proceed as follows. We start off with reviewing related work. Afterwards, we introduce Gesalt Vision (ELVIS). Before concluding, we will present the results of our evaluation using EVLIS.

---

1. https://github.com/ml-research/ELVIS

## 2. Related Work

We will now review the relevant literature focusing on two major subareas, namely visual perception and neuro-symbolic reasoning.

### 2.1. Gestalt Principles and Computer Vision

Gestalt principles have a long and rich history in psychology, tracing back to seminal works by Wertheimer, Koffka, and Palmer (Wertheimer, 1938; Koffka, 1935; Palmer, 1999; Ellis, 1999). In recent decades, these foundational ideas have influenced a variety of computational models in machine learning and computer vision (Lörincz et al., 2017; Hua and Kunda, 2020; Kim et al., 2021; Zhang et al., 2024), often aiming to replicate or approximate the human capacity for grouping and structural organization. Despite these efforts, most research has emphasized convolutional neural networks or other purely neural approaches, leaving a gap in methods that explicitly combine neural perception with symbol-based mechanisms to capture the holistic grouping functions emblematic of Gestalt reasoning.

### 2.2. Neuro-symbolic Learning and Reasoning

Neuro-symbolic approaches have emerged as a prominent paradigm combining neural networks' powerful perception capabilities with symbolic reasoning's interpretability and robustness. A variety of benchmarks have been developed to assess such hybrid systems, notably CLEVR (Johnson et al., 2017), CLEVRER (Yi et al., 2020), and VQA frameworks leveraging ConceptNet and other knowledge graphs (Yi et al., 2018; Mao et al., 2019; Amizadeh et al., 2020; Tan and Bansal, 2019). Despite their advancements, existing benchmarks predominantly focus on lower-level perceptual or question-answering tasks, often limiting their scope to identifying object properties or simple relational reasoning.

Recently, benchmarks exploring more complex relational and abstract reasoning have emerged. Abstract Visual Reasoning (AVR) tasks investigate the generalization of learned concepts to abstract scenarios, challenging models with tasks involving higher-level reasoning and compositional generalization (Hu et al., 2021). For instance, the CLEVR and CLEVRER benchmarks explicitly address reasoning about object interactions and physics-based causal relationships (Yi et al., 2020). Similarly, the Kandinsky Patterns (Müller and Holzinger, 2021) and its extension to 3D scenes (Sha et al., 2024) test model performance on synthetic, structured visual data, highlighting difficulties in understanding abstract relations and grouping concepts. Furthermore, the recent Alphabet-Shape Dataset (Sha et al., 2024), which involves recognizing alphanumeric shapes composed of grouped objects, exemplifies pioneering work explicitly focusing on grouping as a fundamental cognitive principle.

Our work extends these efforts by explicitly incorporating Gestalt principles, thereby significantly deepening the complexity of relational reasoning tasks. By addressing visual grouping phenomena—such as proximity, similarity, etc.—the proposed Gestalt Reasoning Benchmark, ELVIS, rigorously evaluates the capabilities of neuro-symbolic models in interpreting abstract visual patterns. This approach not only enhances the evaluation of symbolic inference but also aligns closely with human perceptual cognition, creating opportunities for future benchmarks and frameworks that better emulate human visual understanding.

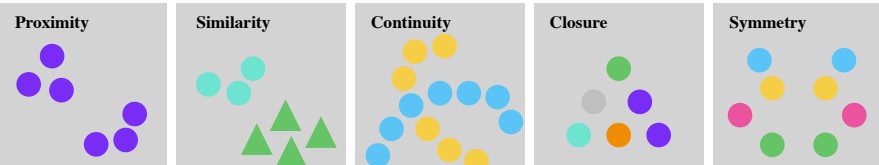

Figure 2: **Gestalt Principles Supported by ELVIS.** From left to right: **Proximity**: Objects that are spatially close to each other are perceived as a group. **Similarity**: Objects with common attributes, such as shape or color, are grouped together. **Continuity**: Objects with continue positions are grouped together. **Feature Closure**: Objects with aligned visual features create an implicit, complete shape. **Position Closure**: Objects arranged in a manner that suggests a closed contour are grouped. **Symmetry**: Objects mirrored across an axis are perceived as a structure, each side determines a group.

## 3. Gesalt Vision (ELVIS): A Gestalt Reasoning Benchmark

Gesalt Vision (ELVIS) is a curated collection of synthetic visual scenes that emphasize five key Gestalt principles: Proximity, Similarity, Closure, Continuity, and Symmetry, as illustrated in Fig. 2. These principles are essential for understanding how discrete visual elements are perceived as cohesive patterns—an important challenge for neuro-symbolic models that integrate learned perceptual features with logical reasoning mechanisms.

### 3.1. Overview of ELVIS

Each task in ELVIS illustrates at least one Gestalt principle by showcasing how objects can be grouped into meaningful units. Rather than merely detecting individual shapes or colors, the benchmark encourages models to identify higher-level relational properties. This approach captures a core facet of visual cognition: elements that share certain features (e.g., proximity, shape similarity, or symmetrical arrangement) tend to be perceived as a unified whole.

By testing the model's ability to detect and interpret these groupings, ELVIS goes beyond basic object recognition. It calls for structured reasoning about the relational organization of objects within a scene—a vital aspect of human-like perception and logical inference. By mimicking human perceptual processing, which tends to generalize well from limited information, machine learning models can learn more robust representations of data. This is especially helpful when training data is scarce, as it allows the model to extrapolate patterns based on high-level Gestalt principles.

### 3.2. Data Generation

ELVIS benchmark is generated under controlled conditions to systematically showcase each Gestalt principle.

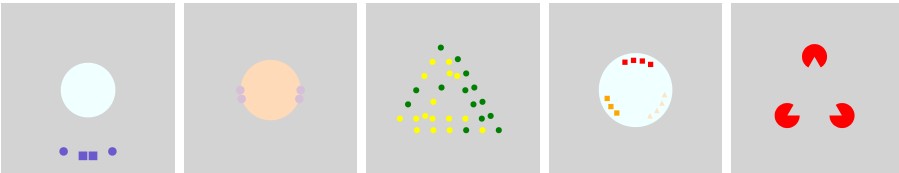

Figure 3: **Geometric Feature Scenarios.** Example patterns illustrating different geometric feature scenarios. **From left to right**: individual objects, object overlap, group overlap, nested shapes, and incomplete forms, designed to assess model perception under varied spatial configurations.

**Diverse Objects.**    Scenes include shapes such as triangles, squares, and circles, varying in color up to 150 variations and spanning a size range of approximately 2% to 80% of the image width. Having diverse objects in the benchmark can make the models more robust and generalizable while also reducing bias and overfitting.

**Varied Complexity.**    Object counts range from just a few in simpler scenes to several hundred in more challenging ones. Regardless of quantity, each scene clearly embodies a target Gestalt principle (*e.g.*, objects arranged to highlight proximity-based clustering). For objects that follow the same Gestalt principle, they can differ widely in shapes, colors, and sizes, which ensures the difficulty arises not only from quantity but also from the heterogeneous attributes of the objects.

**Explicit Groupings.**    Objects are deliberately positioned to leave no ambiguity about the intended grouping cues. This consistency enables more reliable model comparisons by controlling for extraneous factors. Through these design choices, ELVIS aims to push computational models toward context-sensitive reasoning. Instead of limiting models to feature-level classification, they must apply logical rules to recognized shapes, colors, and sizes to determine how individual elements come together into coherent, interpretable wholes. These capabilities are essential for neuro-symbolic systems striving to bridge the gap between pixel-level perception and symbolic-level reasoning.

### 3.3. Features in the patterns

Although the patterns are composed of basic geometric shapes (*i.e.*, triangle, square, and circle), their variations extend beyond simple shape detection. Fig. 3 illustrates five distinct scenarios designed to challenge the robustness of perception models: **Individual**, the objects are placed individually without overlapping, which is the most straightforward case; **Object Overlap**, the objects are overlapped with each other, which can cover part of the features of some of the objects in the image; **Group Overlap**, multiple groups are overlapped with each other, whereas the objects are still remaining individual. **Inside**, the objects are completely inside another object; and **Incomplete**, the object is not completely drawn in the image, which sometimes shows the features of other shapes.

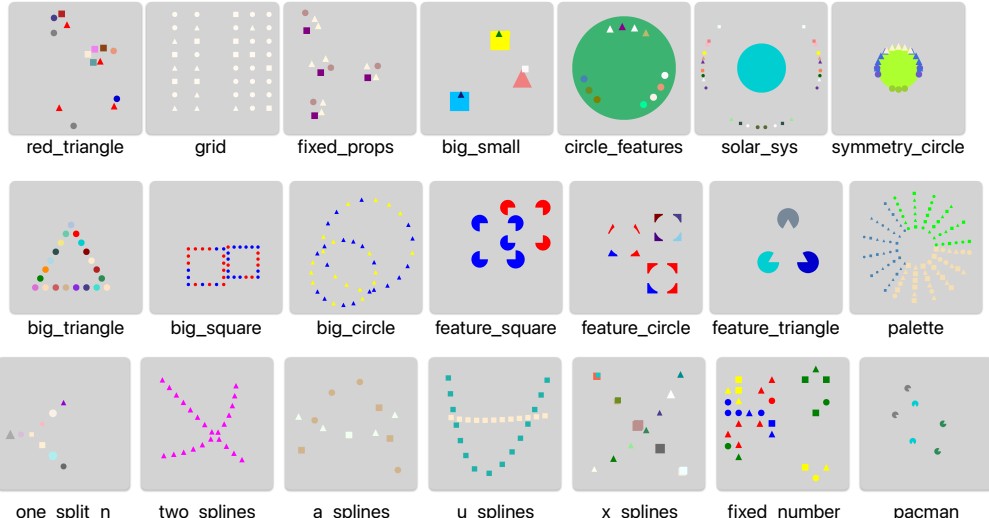

Figure 4: **Category Base Patterns of ELVIS**. Each category in ELVIS is based on a specific Gestalt principle. The base pattern of each category serves as a foundational structure, which can generate numerous variations by adjusting object properties.

These variations test the model's ability to handle occlusion, containment, and missing features, ensuring a deeper understanding of geometric properties.

### 3.4. Category

Although we provide hundreds of tasks for each Gestalt principle, we do not treat them as entirely separate scenarios. Instead, we introduce a base pattern called a *category* to efficiently generate multiple tasks. Each category is explicitly designed around a specific Gestalt principle. By modifying key attributes, such as the number of groups, the number of objects within each group, and the color, shape, or size of each object, we can create numerous variations while maintaining the same underlying principle. Fig. 4 presents examples of each category used in the ELVIS.

### 3.5. Task Formulation

Each task in ELVIS is defined by a set of rules, which specifies a combination of logical conditions that determine the structure of valid visual patterns. These rules are instantiated as constraints on object-level properties (e.g., shape, color, size) and group-level configurations (e.g., spatial arrangement, symmetry). For example, a rule might require that each group contains one red triangle, or several objects form a symmetrical structure.

Using these rules, the dataset generation pipeline creates a set of positive images that fully satisfy all constraints and a corresponding set of negative images, each of which violates at least one constraint. Each image is assigned a binary label: 1 for a positive sample and

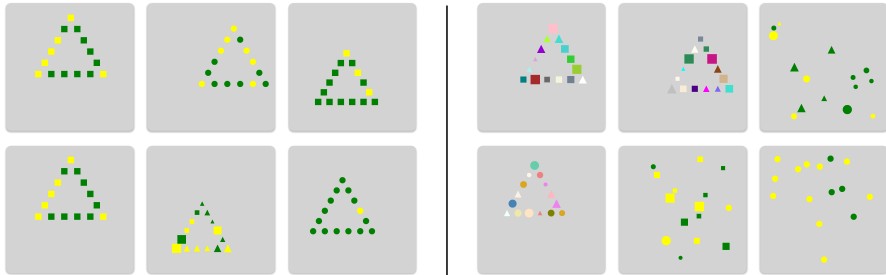

Figure 5: **Task Example of ELVIS**. **Left**: positive patterns following the Gestalt principle of closure, forming a yellow-green triangle. **Right**: negative patterns that partially follow the rules but violate key logic constraints.

0 for a negative one. A task is defined as the classification problem of distinguishing these two types of images based on their compliance with the underlying rules.

In this setting, the rules capture the complete logical structure of the visual pattern, the constraints represent the atomic predicates that compose the rules, the label indicates whether an image satisfies all constraints, and the task refers to the binary classification challenge associated with rules. Although some negative images may share superficial similarities with positive ones, they are guaranteed to break at least one essential constraint, making the task nontrivial and requiring more than low-level visual matching.

This formulation allows models to be evaluated in a focused and interpretable manner, testing their ability to infer meaningful group-level properties from structured visual input. While ELVIS is primarily constructed for binary classification over paired image sets, it is flexible enough to support other task types, such as single-image classification, Gestalt principle prediction, and pattern completion. A discussion of these alternative task modes and their potential applications is provided in Appendix A.

## 4. Empirical Evaluation using ELVIS

We now evaluate the ELVIS benchmark with some state-of-the-art neural and neuro-symbolic methods to demonstrate the shortcoming(s) of current machine learning models.

### 4.1. Task Types and Evaluation Metrics

ELVIS comprises a diverse set of tasks designed to evaluate how effectively computational models can identify and reason about Gestalt principles. Tab. 1 summarizes the task distribution. Each principle is associated with hundreds of tasks that feature considerable variation in visual complexity, such as object count (ranging from a few to several hundred), color diversity (hundreds of different colors), object shapes (limited to three for controlled variability), and object sizes (varying between 2% and 80% of the width of the image). These variations ensure that the benchmark tests a wide array of perceptual scenarios.

Models were trained and evaluated independently for each task. Specifically, for a given task, a model was trained on its corresponding labeled examples and evaluated on its own

Table 1: **Benchmark tasks summarization** The table summarizes the benchmark tasks for each Gestalt principle. Columns from left to right indicate principles, category number, task number, object number, group number, color number, shape number, and the object size range.

| Principle | # Cat. | # Task | # Obj. | # G. | # Color | # Shape | Size Range |
|---|---|---|---|---|---|---|---|
| Proximity | 5 | 234 | 4-64 | 2-4 | 150 | 3 | $5\% \sim 80\%$ |
| Similarity | 3 | 183 | 4-162 | 1-4 | 150 | 3 | $2\% \sim 10\%$ |
| Closure | 6 | 171 | 3-60 | 1-4 | 150 | 3 | $3\% \sim 12\%$ |
| Continuity | 2 | 225 | 6-45 | 2-3 | 150 | 3 | $3\% \sim 8\%$ |
| Symmetry | 5 | 120 | 5-25 | 1-4 | 150 | 3 | $5\% \sim 40\%$ |

Table 2: **Large Models Comparison** Two large models were used for benchmark evaluation. The ViT refers ViT-B/16 model, which is a vision transformer pretrained on ImageNet-1K, while LlaVA refers LLaVA-OneVision-Qwen2-7B-SI, which is a multi-modal model incorporating text-image understanding.

| Model | Pretrain. D. | Img. R. | Params(M) | FLOPs(B) |
|---|---|---|---|---|
| ViT | ImageNet-1K | $224 \times 224$ | 86 | 17.6 |
| LlaVA | Multi-modal | $224 \times 224$ | 7000 | 1200+ |
| NEUMANN | N/A | $224 \times 224$ | $\approx 0$ | $\approx 0$ |

held-out test set. This process was repeated separately for every task in the benchmark. For each task, we report performance using both accuracy and F1 score. The latter provides a more balanced view by accounting for precision and recall. The final benchmark performance is then computed as the mean and standard deviation of these metrics across all tasks.

### 4.2. Baseline Models

We evaluated three representative baselines, encompassing neural and neuro-symbolic approaches. Tab. 2 summarizes the characteristics of the baseline models.

**Vision Transformer (ViT-B/16)** (Wu et al., 2020; Wightman, 2019) is a purely neural model pre-trained on ImageNet-21K and fine-tuned on ImageNet-1K, providing strong visual perception capabilities at a resolution of $224 \times 224$ pixels. It is a transformer-based vision model that represents an image as a sequence of patch tokens rather than using convolutional features. Each image is split into $16 \times 16$ patches which are embedded into vector tokens. **NEUMANN** (Shindo et al., 2024) is a neuro-symbolic hybrid model that integrates learned neural perception with symbolic logic-based reasoning. It introduces a form of differentiable logic programming to tackle abstract visual reasoning problems that go beyond straightforward perception. More details of the baseline are in Appendix C. **LLaVA-OneVision** (Li et al., 2024), an advanced multimodal Large Language Model (LLM) that extends text-based language modeling to incorporate visual inputs. Built upon the Qwen2 LLM as its language backbone, LLaVA-OneVision is fine-tuned on extensive multimodal instruction data—for example, image-question-answer pairs and vision-language dialogues.

Table 3: **Performance Comparison.** The table presents the average score and standard deviation across four metrics: accuracy, F1 score, precision, and recall. ViT = ViT-B/16 model, Llava = LlaVA-OneVision-Qwen2-7B-SI, and NM = NEUMANN.

| Met. | Model | Proximity | Similarity | Closure | Symmetry | Continuity |
|------|-------|-----------|------------|---------|----------|------------|
| Acc. | ViT/3 | $0.52 \pm 0.22$ | $0.52 \pm 0.22$ | $0.54 \pm 0.18$ | $0.50 \pm 0.14$ | $0.57 \pm 0.17$ |
| | ViT/100 | $0.50 \pm 0.04$ | $\mathbf{0.60} \pm 0.15$ | $0.60 \pm 0.11$ | $0.50 \pm 0.04$ | $\mathbf{0.68} \pm 0.17$ |
| | Llava/3 | $0.50 \pm 0.11$ | $0.50 \pm 0.09$ | $\mathbf{0.62} \pm 0.16$ | $\mathbf{0.63} \pm 0.19$ | $0.54 \pm 0.12$ |
| | NM/3 | $\mathbf{0.53} \pm 0.16$ | $0.51 \pm 0.08$ | $0.57 \pm 0.15$ | $0.49 \pm 0.09$ | $0.53 \pm 0.13$ |
| F1 | ViT/3 | $0.40 \pm 0.35$ | $0.37 \pm 0.35$ | $0.48 \pm 0.31$ | $0.41 \pm 0.28$ | $0.52 \pm 0.30$ |
| | ViT/100 | $0.00 \pm 0.02$ | $\mathbf{0.51} \pm 0.32$ | $\mathbf{0.57} \pm 0.23$ | $0.02 \pm 0.02$ | $\mathbf{0.65} \pm 0.24$ |
| | Llava/3 | $\mathbf{0.40} \pm 0.14$ | $0.36 \pm 0.10$ | $0.53 \pm 0.22$ | $\mathbf{0.58} \pm 0.22$ | $0.44 \pm 0.16$ |
| | NM/3 | $0.20 \pm 0.30$ | $0.20 \pm 0.25$ | $0.42 \pm 0.31$ | $0.15 \pm 0.22$ | $0.25 \pm 0.28$ |
| Pre. | ViT/3 | $\mathbf{0.35} \pm 0.33$ | $0.31 \pm 0.33$ | $0.46 \pm 0.31$ | $0.41 \pm 0.31$ | $0.48 \pm 0.31$ |
| | ViT/100 | $0.00 \pm 0.04$ | $0.49 \pm 0.31$ | $\mathbf{0.57} \pm 0.22$ | $0.01 \pm 0.08$ | $\mathbf{0.69} \pm 0.21$ |
| | Llava/3 | $0.26 \pm 0.35$ | $0.22 \pm 0.28$ | $0.49 \pm 0.29$ | $\mathbf{0.63} \pm 0.30$ | $0.32 \pm 0.41$ |
| | NM/3 | $0.30 \pm 0.41$ | $0.21 \pm 0.36$ | $0.45 \pm 0.34$ | $0.32 \pm 0.39$ | $0.34 \pm 0.39$ |
| Rec. | ViT/3 | $\mathbf{0.54} \pm 0.47$ | $0.52 \pm 0.48$ | $0.59 \pm 0.41$ | $0.49 \pm 0.38$ | $0.62 \pm 0.40$ |
| | ViT/100 | $0.01 \pm 0.07$ | $\mathbf{0.61} \pm 0.41$ | $0.66 \pm 0.32$ | $0.03 \pm 0.16$ | $\mathbf{0.69} \pm 0.29$ |
| | Llava/3 | $0.26 \pm 0.36$ | $0.35 \pm 0.46$ | $\mathbf{0.73} \pm 0.40$ | $\mathbf{0.71} \pm 0.35$ | $0.23 \pm 0.32$ |
| | NM/3 | $0.18 \pm 0.30$ | $0.14 \pm 0.23$ | $0.42 \pm 0.35$ | $0.16 \pm 0.22$ | $0.21 \pm 0.26$ |

## 4.3. Quantitative Evaluation

Tab. 3 and Fig. 6 summarize the the performance comparisons of the three baseline models across five Gestalt principles. Overall, the mean accuracy typically hovers near 50%, indicating that these tasks pose a significant challenge even for state-of-the-art neural and neuro-symbolic systems. F1 scores reveal a pattern of relatively higher recall than precision across all methods, suggesting a bias toward over-identifying positives.

A notable trend is that increasing the training set size (ViT-B/100 vs. ViT-B/3) often improves F1 scores on select categories such as `palette` and `x_splines`, highlighting the importance of substantial visual supervision for robust feature extraction. Nonetheless, the gains are inconsistent: despite observing improvements on continuity and closure, no clear advantage emerges for principles like proximity or symmetry. This discrepancy points to the possibility that certain visual cues (e.g. spacing or symmetrical structures) may be more elusive for a purely neural encoder, even with additional data.

LLaVA, in contrast, often achieves competitive or superior results despite limited training data, underscoring the effectiveness of its multimodal instruction tuning for few-shot learning. Its strengths on `closure` and `symmetry` exemplify how a large language model backbone, guided by suitable multimodal alignments, can compensate for small class-specific datasets. Meanwhile, NEUMANN's overall weaker performance highlights a common neuro-symbolic bottleneck: while symbolic inference offers powerful reasoning capabilities, imperfect perceptual grounding can derail the entire pipeline. These findings collectively underscore the interplay between data sufficiency, perceptual modeling, and high-level reasoning, which remains a core challenge in neuro-symbolic learning.

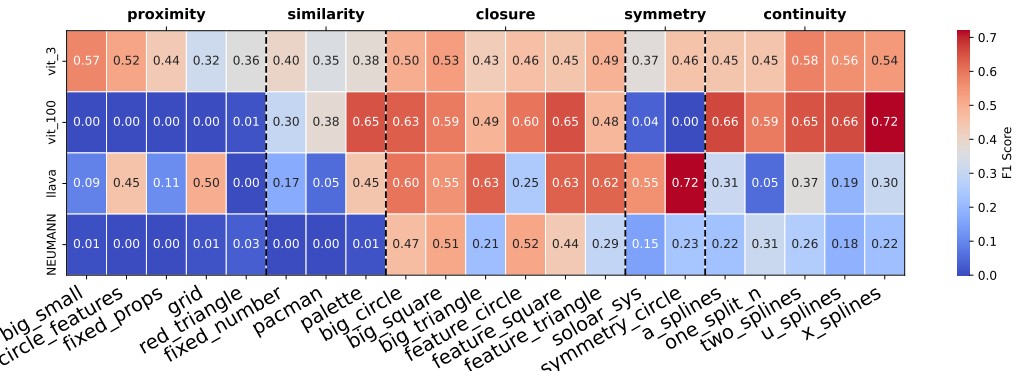

Figure 6: **Average F1 score by Categories Over baseline models**. The chart compares average F1 scores (y-axis) for proximity, similarity, closure, symmetry, continuity, and related categories (x-axis). Larger training sets (ViT/100) help on some categories (e.g., `palette`, `x_splines`), yet LLaVA remains competitive with limited data, while NEUMANN underscores the perceptual bottleneck in neuro-symbolic reasoning.

## 4.4. Limitations and Insights

ELVIS inherently contains biases from synthetic image generation, potentially limiting generalizability to real-world scenarios. Additionally, simplified object shapes and discrete principle-based patterns, while facilitating controlled experimentation, might not fully capture the complexity of natural visual cognition. Models exhibiting high variance in performance across different Gestalt principles suggest opportunities for further optimization and deeper integration of symbolic reasoning with advanced perceptual models.

## 5. Conclusion and Future Work

We introduced the Gestalt Vision (ELVIS) benchmark, a benchmark specifically tailored to evaluate neuro-symbolic systems on five core Gestalt principles: Proximity, Similarity, Closure, Continuity, and Symmetry. By designing visual tasks that focus on relational properties, ELVIS challenges models to move beyond basic object detection and engage in higher-level logical reasoning. The methodology of ELVIS involves generating tasks from base categories, ensuring scalability and systematic variation in object count, shape, color, and size. Through comparative evaluations of multiple baseline approaches, we observed that overall performance often hovers around chance level, reflecting the nuanced nature of these tasks. While neural models benefit from larger datasets, neuro-symbolic methods underscore the importance of integrating perceptual accuracy with logical inference.

Future research could enhance synthetic scene realism to bridge the gap to natural images. Improving neuro-symbolic frameworks for better precision-recall balance may also boost generalization. Ultimately, ELVIS aims to be both a diagnostic tool and a catalyst for advancing perceptual reasoning systems.

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

## Appendix A. Alternative Task Modes

Although ELVIS primarily supports binary classification over grouped image sets (positive vs. negative), its design allows alternative usage modes:

- Single-image prediction: For each task, a classifier may be trained to label individual images as satisfying the unknown rule (positive) or not.

- Self-supervised pattern completion: Some categories (e.g., closure or symmetry) can support tasks where part of the scene is missing and must be inferred.

- Principle inference: Given a scene, predict which Gestalt principle it expresses (e.g., proximity, closure). While not yet benchmarked, ELVIS supports this via task metadata.

These extensions are intended to be supported in future releases. A task loader and dataset API in the style of CLEVR's SceneGraph API is also under development.

## Appendix B. Task Examples

For each Gestalt principle in ELVIS, we present one or two representative task categories to illustrate the underlying design. The category names serve as intuitive references, but they do not always reflect the full range of variations. Due to controlled perturbations, some task variants may differ significantly from their original category name.

For instance, the category `Red Triangle` is initially designed around the idea that each group contains one red triangle. However, certain variations derived from this category may disregard color in the rule, resulting in tasks where the correct answer is determined solely by the presence of a triangle—regardless of its color. These variants are still formally associated with the `Red Triangle` category, though their governing logic differs. Other categories follow the same behavior.

### B.1. Proximity: Red Triangle

The pattern `Red Triangle` follows the Gestalt principle of proximity. The base pattern is structured with multiple object groups, where each group consists of at least one red triangle and several smaller ones placed closely together.

Fig. 7 presents a task where the rule is defined by *color* and *shape*. In the positive pattern, each group contains at least one object with red color and triangle shape, with the rest being random properties.

Fig. 8 illustrates another task variation, incorporating *color* only. In the positive pattern, each group contains at least one red object; the shape of the red object is randomly determined.

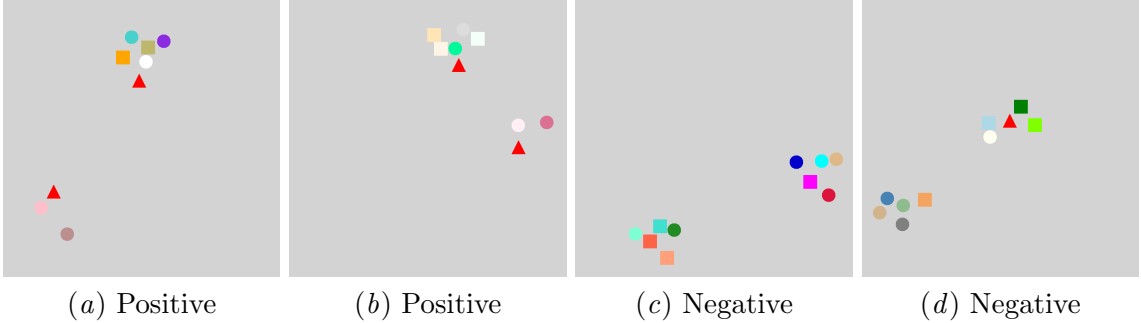

$(a)$ Positive  $(b)$ Positive  $(c)$ Negative  $(d)$ Negative

Figure 7: Red Triangle: Considering shape, and color into the logic rules.

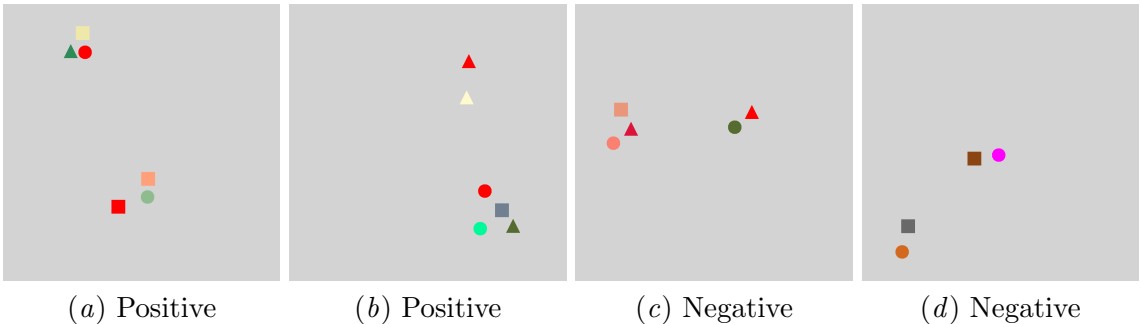

$(a)$ Positive $\qquad$ $(b)$ Positive $\qquad$ $(c)$ Negative $\qquad$ $(d)$ Negative

Figure 8: Red Triangle: Considering color into the logic rules.

## B.2. Proximity: Big Small

The pattern `Big Small` follows the Gestalt principle of proximity. The base pattern is structured with multiple object groups, where each group consists of one large object and several smaller ones placed closely together.

Fig. 9 presents a task where the rule is defined by *count* and *size*. In the positive pattern, each group contains exactly one large object, with the rest being small, and there are precisely three groups.

Fig. 10 illustrates another task variation, incorporating *color* and *size*. In the positive pattern, each group follows the same size structure, but object colors are restricted to either green or yellow.

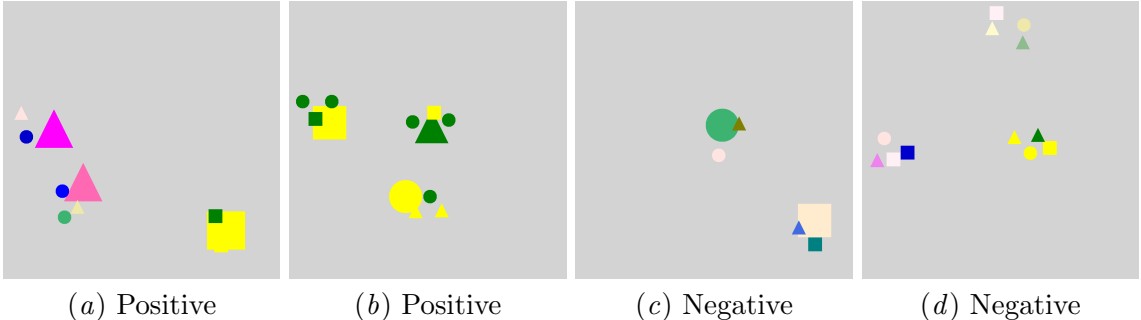

$(a)$ Positive $\qquad$ $(b)$ Positive $\qquad$ $(c)$ Negative $\qquad$ $(d)$ Negative

Figure 9: Big Small: Considering count, and size into the logic rules.

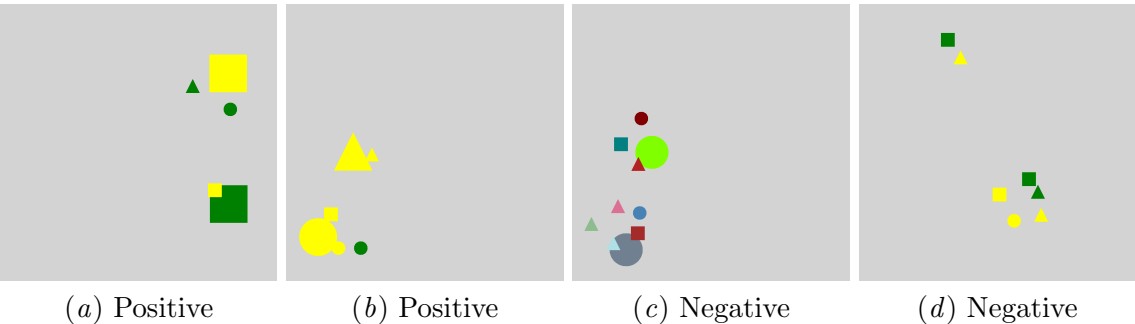

(a) Positive  (b) Positive  (c) Negative  (d) Negative

Figure 10: Big Small: Considering color, and size into the logic rules.

## B.3. Similarity: Fixed Number

The category `Fixed Number` is based on the Gestalt principle of similarity. The base pattern consists of an equal number of objects in different colors, with up to four color variations. Additionally, object size and shape can vary to introduce further task variations.

Fig. 11 illustrates a task where the rule involves counting objects of two colors.

Fig. 12 presents a variation where the task requires counting objects among four colors.

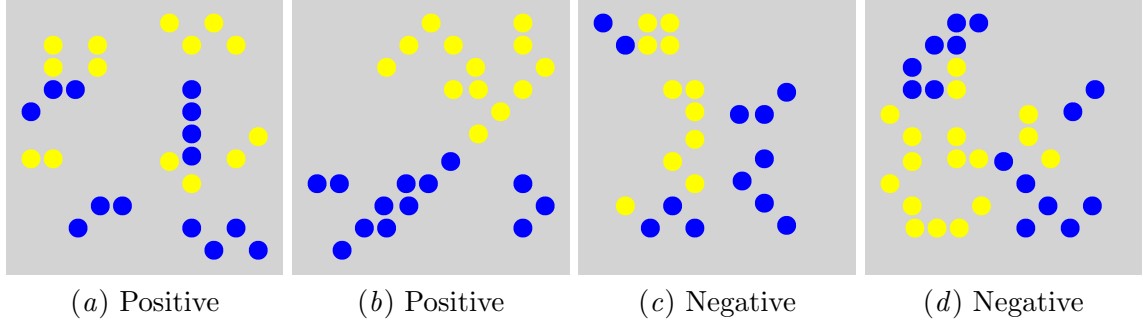

(a) Positive  (b) Positive  (c) Negative  (d) Negative

Figure 11: Fixed Number: 2 Colors

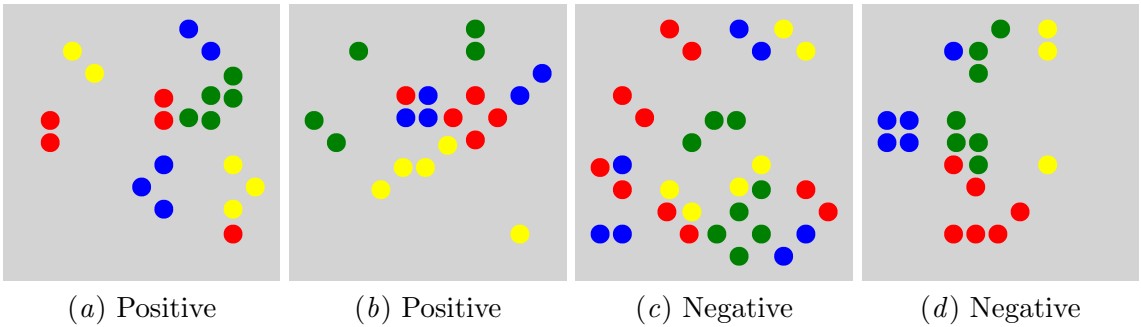

(a) Positive  (b) Positive  (c) Negative  (d) Negative

Figure 12: Fixed Number: 4 Colors

### B.4. Closure: Feature Square

The category `Feature Square` follows the Gestalt principle of closure. Its base pattern consists of four 3/4 circles arranged to outline a square. Fig. 13 illustrates a task where object colors are limited to red or blue. Fig. 14 presents a variation where all circles are of equal size. Each task includes a counterfactual pattern that disrupts closure while maintaining all other rules.

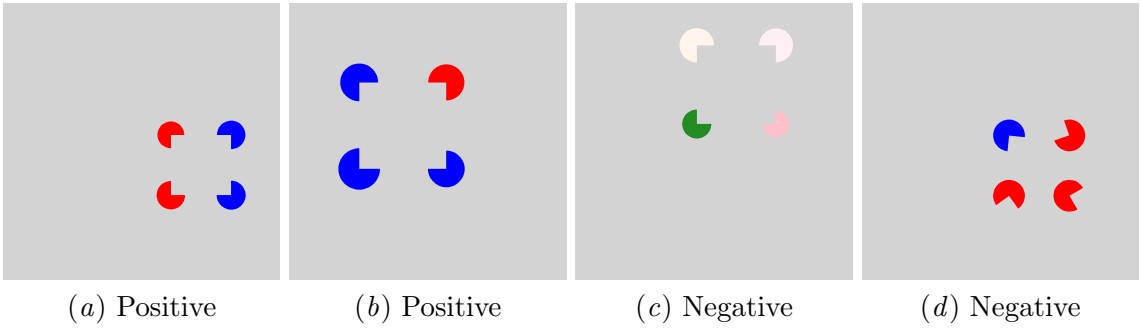

(a) Positive      (b) Positive      (c) Negative      (d) Negative

Figure 13: Feature Square: Color

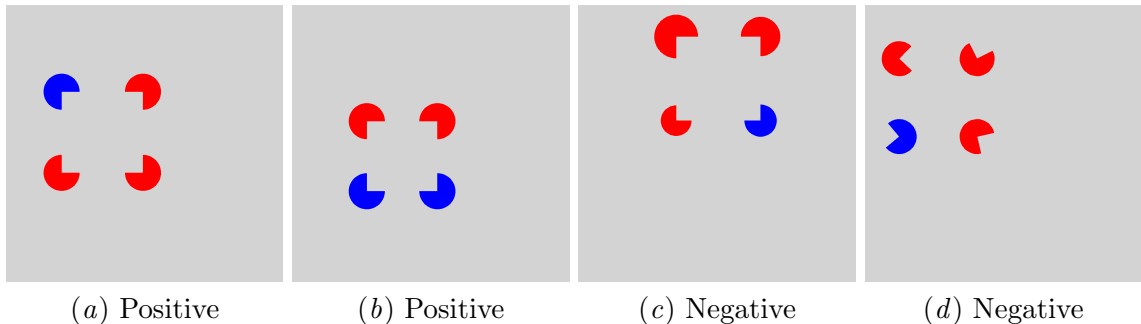

(a) Positive      (b) Positive      (c) Negative      (d) Negative

Figure 14: Feature Square: Size

### B.5. Symmetry: Solar Sys

The category `Solar Sys` follows the Gestalt principle of symmetry. Its base pattern consists of a large central circle with smaller objects symmetrically positioned around it.

Fig. 15 illustrates a task where small object shapes are limited one variation per image. Fig. 16 presents a variation where both colors and shapes are restricted to at most two variations.

### B.6. Continuity: Two Splines

The category `Two Splines` follows the Gestalt principle of continuity. Its base pattern consists of two intersecting splines formed by small objects.

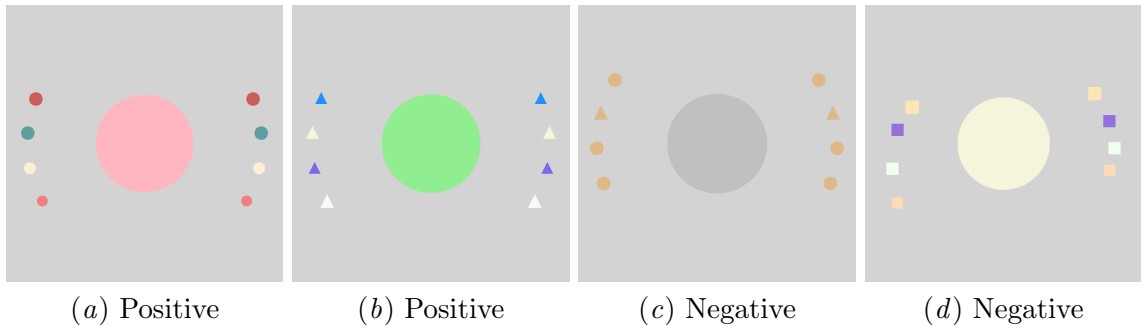

(*a*) Positive     (*b*) Positive     (*c*) Negative     (*d*) Negative

Figure 15: Solar Sys: Shape

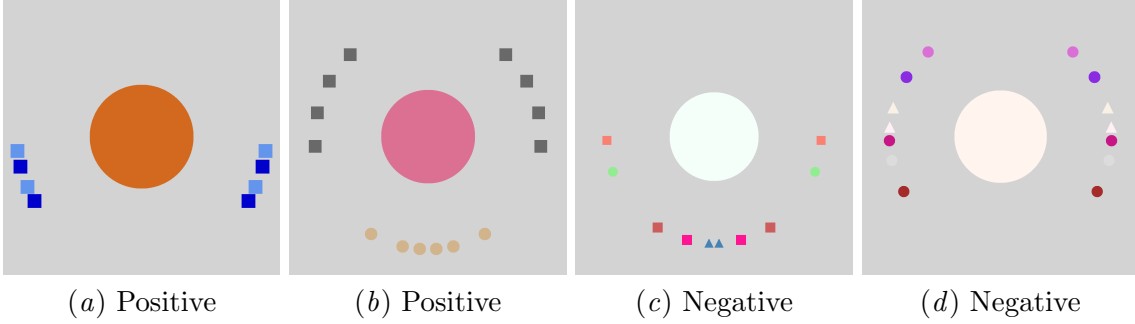

(*a*) Positive     (*b*) Positive     (*c*) Negative     (*d*) Negative

Figure 16: Solar Sys: Color, Shape

Fig. 17 illustrates a task where all objects share the same shape. Fig. 18 presents a variation where both the colors and shapes of the objects are identical.

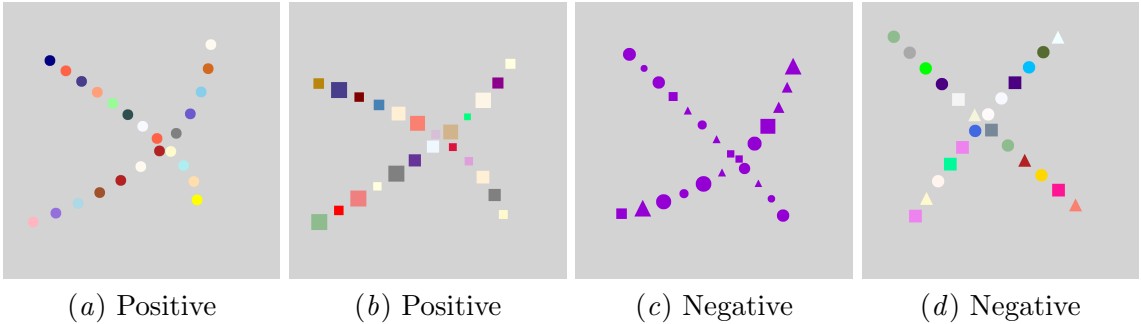

(*a*) Positive   (*b*) Positive   (*c*) Negative   (*d*) Negative

Figure 17: Two Splines: Shape

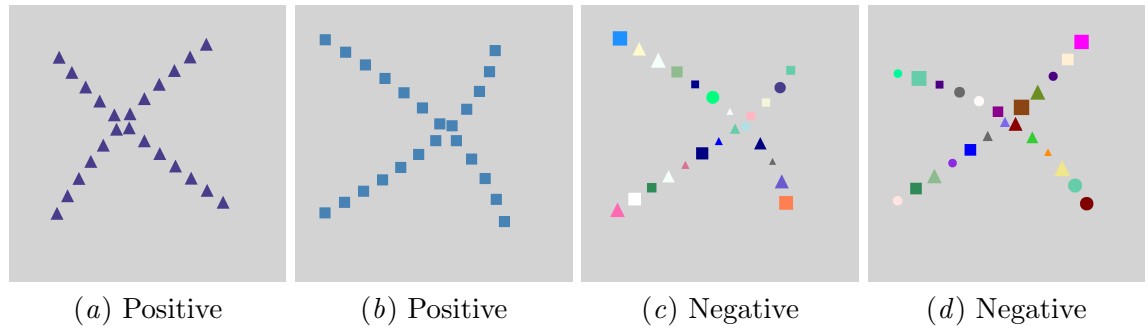

(*a*) Positive   (*b*) Positive   (*c*) Negative   (*d*) Negative

Figure 18: Two Splines: Color, Shape

## Appendix C. Experimental Details

We followed the setup in the paper (Shindo et al., 2024) to train the NEUMANN baseline. We used the public YOLOv5[2] model. We adopted the YOLOv5s model, which has 7.3M parameters. The model is pre-trained with $15,000$ pattern-free figures for training, 5000 figures for validation. The class labels and positions are generated randomly. The label consists of the class labels and the bounding box for each object. The class label is generated by the combination of the shape and the color of the object, e.g., *red circle* and *blue square*. We trained the NEUMANN model for 100 epochs with a batch size of 64. We used the RMSProp (Ruder, 2016) optimizer with a learning rate of $1e-2$.

---

2. https://github.com/ultralytics/yolov5

