# OpenReview forum: "Gestalt Vision: A Dataset for Evaluating Gestalt Principles in Visual Perception"
_nesyconf.org/NeSy/2025/Conference — NeSy 2025 Poster_

### Official Review · Reviewer_qKLT · 2025-04-01
**Filling the gap between low-level perception and symbolic reasoning**

**Rating:** 9
**Confidence:** 5

**Review:**

This paper offers a compelling and distinctive perspective on cognitive reasoning by leveraging principles from Gestalt theory. The authors argue convincingly that contemporary machine vision algorithms, which predominantly emphasize object-level visual classification, fail to adequately capture low-level perceptual attributes such as contour details and shading. These features, extensively examined by early 20th-century psychologists and philosophers, remain largely overlooked in current methods. This oversight similarly affects symbolic reasoning tasks, where reasoners typically receive object categories and trajectories without detailed low-level perceptual inputs.

To substantiate their claims, the authors introduce ELVIS, a synthetic dataset specifically designed to highlight fundamental shapes and Gestalt-driven perceptual phenomena. Evaluations demonstrate that existing state-of-the-art models generally exhibit poor performance on this novel benchmark, underscoring the authors' argument effectively.

Overall, the paper is original, well-written, and highly suitable for presentation at the NeSy conference.

Additional comments:
- While ELVIS is presented as a "diagnostic framework," the paper does not clearly delineate a novel diagnostic methodology beyond the creation of the ELVIS dataset itself. It appears that the primary contribution is the dataset, followed by standard evaluations. Clarification on this point would enhance the manuscript.
- Section 4 (Evaluation) could benefit from a broader evaluation, possibly including additional relevant models such as CLIP to provide a more comprehensive comparative analysis.
- Section 5 raises an intriguing possibility for future work. Beyond Gestalt theory, are the authors considering extending their investigation to incorporate conceptual schemas or frameworks referenced in related literature?


Langacker, R. W. (2002). Concept, image, and symbol: The cognitive basis of grammar (2nd ed.). Berlin: Mouton de Gruyter.
L. Talmy. Toward a Cognitive Semantics. vol. 1, Cambridge (Massachusetts)

**Anonymity:**

Remain anonymous

---

### Official Review · Reviewer_YuAx · 2025-04-03
**An attempt to propose a synthetic benchmark for learning and assessment of capturing of Gestalt principles**

**Rating:** 4
**Confidence:** 5

**Review:**

This paper proposes a visual benchmark that revolves around the Gestalt principles, and is meant to facilitate training of models that should capture those principles.

The paper is quite well written and the motivations behind this project are well grounded. However, there are several weak points:
1. The proposed benchmark seems to focus exclusively on the Gestalt principles, so that tasks examine only those principles in learning agents. As a matter of fact, vision is not only about Gestalt -- there are many other characteristics of visual perception that are essential for robust scene interpretation, and Gestalt is only part of them.

2. The benchmark feels quite 'closed' and narrowly-targeted, so to say: the tasks are posed quite specifically in terms of identifying the logical constraints that distinguish two groups of images (Fig. 5 and Sec. 3.5). A well constructed benchmark should offer more flexibility and multiple usage scenarios, e.g. prediction of the principles that hold in a given single image, completion of an 'unfinished' visual pattern (in the spirit of self-supervised learning), and more. This could be facilitated by offering some sort of application programming interface to the benchmark (in a similar way to the one provided by the authors of CLEVR, for that instance). Unfortunately, it does not seem that the authors offer such an option -- no mention of that in the main text nor in the appendix.

Overall, a serious proposal of a new benchmark should promise a number of convenient usage options. While I understand that the authors could not for instance provide a URL to their github project for the reasons of anonymity, much more could and should have been said in this submission anyway.

3. The description of the benchmark seems to be incomplete: a number of conceptual and technical assumptions are missing (while they could have been easily conveyed at least in the appendix). For instance, in the last paragraph of Sec. 3.5, the authors state:

> To solve the task, a model must identify all the logical constraints that hold true in the positive samples and do not apply to the negative ones.

But what does 'identify' mean in this context? My guess is that we're talking here in terms of posing the task as multi-labelled classification, i.e. a given task may have an arbitrary number of constraints involved in it, and the task of the model is to predict them correctly. But this causes a technical issue: a given principle can be instantiated in a given task in a number of ways, e.g. Similarity may concern any visual property, like shape, color, size, etc. So the number of classes/tags in the above sense would be combinatorial, quite large, which feels inconvenient.

Put more technically, the manuscript failed to specify more technically what is the 'target' (the 'desired output') of the agents that attempt to solve these tasks.

4. Last but not least, the paper is too informal at quite many places: the concepts like 'rule', 'constraint', 'label' are not given sound definitions in the text.


Editorial remarks:

There are quite significant repetitions in the text; some arguments brought up at the very beginning of the paper are repeated several times until page 4.

'alphabetic shape'
- somewhat imprecise phrase; do you mean 'alphanumeric symbols'?

**Anonymity:**

Remain anonymous

---

### Official Review · Reviewer_tQeX · 2025-04-04
**new visual reasoning benchmark**

**Rating:** 7
**Confidence:** 4

**Review:**

The manuscript proposes a new dataset for visual perception and reasoning based on Gestalt principles. It is based on a series of synthetic datasets of positive and negative examples - to distinguish them, the solver must identify all the logical constraints that hold true in the
positive samples and do not apply to the negative ones.

Many visual reasoning benchmarks are available, but they are often based on the relationships between objects and their properties. The present dataset focuses on a systematic exploration of Gestalt principles, that pertain to how human organize visual information into patterns and wholes, including principles like similarity, proximity, continuity, closure, and symmetry. Thus, this type of datasets stresses the combination of high-level semantic image interpretation (the fact that there are four triangles) and low-level visual patterns (the fact that the four triangles form and are perceived as a rectangle). As such, it is conceivable that such benchmarks may stress current neuro-symbolic AI techniques that rely on an efficient and effective mapping between the symbolic and the sub-symbolic.

Strenghts:
- interesting benchmark for the Nesy and visual reasoning community, comprehensive and organized according to different principles
- experimental comparison of both LLMs and neuro-symbolic techniques shows the challenge of the task

Weaknesses:
- synthetic, 2D dataset may not well predict real-life performance
- the authors do not provide a standardize training/validation/test split.
- given that the dataset is the main contribution, and the complexity of the benchmark itself, it would be beneficial that the dataset is made public upon acceptance
- experimental comparison may be biased towards the LLM, which are pretrained, whereas the neuro-symbolic model is trained directly on the dataset

Additional remarks:
- It would be useful to add a more extensive comparison, perhaps in table format, with respect to existing benchmarks
- It is not clear how performance are evaluated. My understanding is that evaluation is performed by averaging on the tasks, but I am not sure I understand correctly. For proximity, for example, there are 234 task. Each task, if I understand correctly, is supported by several images (how many?) Performance in Table 3, according to this interpretation, should be the mean and standard deviation across all tasks. However, is a single model trained on all tasks combined, or this the average of 234 models? In that is so, running this benchmark could be computationally very intensive.
- Regarding NEUMANN, more details should be given on how it was trained, especially how the input image is mapped to probabilistic atoms.

**Anonymity:**

Remain anonymous